Development and evaluation of rapid novel isothermal amplification assays for important veterinary pathogens: Chlamydia psittaci and Chlamydia pecorum

Jelocnik Martina Martina.jelocnik@research.usc.edu.au 1
Islam Md. Mominul 1
Madden Danielle 1
Jenkins Cheryl 2
Branley James 3
Carver Scott 4
Polkinghorne Adam 1
1 Centre for Animal Health Innovation, University of the Sunshine Coast , Maroochydore , Queensland , Australia
2 NSW Department of Primary Industries, Elizabeth Macarthur Agricultural Institute , Menangle , New South Wales , Australia
3 Nepean Hospital , Penrith , New South Wales , Australia
4 School of Biological Sciences, University of Tasmania , Hobart , Tasmania , Australia
Gillespie Joseph
Electronic publication date: 2017 Sep 8
Publication date: 2017
Volume: 5
Electronic Location ID: e3799
Received 2017 Jul 20; Accepted 2017 Aug 22
Copyright: ©2017 Jelocnik et al.
Copyright year: 2017
Copyright holder: Jelocnik et al.
License: This is an open access article distributed under the terms of the Creative Commons Attribution License, which permits unrestricted use, distribution, reproduction and adaptation in any medium and for any purpose provided that it is properly attributed. For attribution, the original author(s), title, publication source (PeerJ) and either DOI or URL of the article must be cited.
License URL: https://creativecommons.org/licenses/by/4.0/

Keywords: Chlamydia psittaci, Chlamydia pecorum, LAMP, Diagnostics, Rapid tests, Clinical samples

Funding: University of the Sunshine Coast Research Seed This work was funded by the University of the Sunshine Coast Research Seed Grant awarded to Martina Jelocnik. The funders had no role in study design, data collection and analysis, decision to publish, or preparation of the manuscript.

==============================
Background

Chlamydia psittaci and Chlamydia pecorum are important veterinary pathogens, with the former also being responsible for zoonoses, and the latter adversely affecting koala populations in Australia and livestock globally. The rapid detection of these organisms is still challenging, particularly at the point-of-care (POC). In the present study, we developed and evaluated rapid, sensitive and robust C. psittaci-specific and C. pecorum-specific Loop Mediated Isothermal Amplification (LAMP) assays for detection of these pathogens.

Methods and Materials

The LAMP assays, performed in a Genie III real-time fluorometer, targeted a 263 bp region of the C. psittaci-specific Cps_0607 gene or a 209 bp region of a C. pecorum-specific conserved gene CpecG_0573, and were evaluated using a range of samples previously screened using species-specific quantitative PCRs (qPCRs). Species-specificity for C. psittaci and C. pecorum LAMP targets was tested against DNA samples from related chlamydial species and a range of other bacteria. In order to evaluate pathogen detection in clinical samples, C. psittaci LAMP was evaluated using a total of 26 DNA extracts from clinical samples from equine and avian hosts, while for C. pecorum LAMP, we tested a total of 63 DNA extracts from clinical samples from koala, sheep and cattle hosts. A subset of 36 C. pecorum samples was also tested in a thermal cycler (instead of a real-time fluorometer) using newly developed LAMP and results were determined as an end point detection. We also evaluated rapid swab processing (without DNA extraction) to assess the robustness of these assays.

Results

Both LAMP assays were demonstrated to species-specific, highly reproducible and to be able to detect as little as 10 genome copy number/reaction, with a mean amplification time of 14 and 24 min for C. psittaci and C. pecorum, respectively. When testing clinical samples, the overall congruence between the newly developed LAMP assays and qPCR was 92.3% for C. psittaci (91.7% sensitivity and 92.9% specificity); and 84.1% for C. pecorum (90.6% sensitivity and 77.4% specificity). For a subset of 36 C. pecorum samples tested in a thermal cycler using newly developed LAMP, we observed 34/36 (94.4%) samples result being congruent between LAMP performed in fluorometer and in thermal cycler. Rapid swab processing method evaluated in this study also allows for chlamydial DNA detection using LAMP.

Discussion

In this study, we describe the development of novel, rapid and robust C. psittaci-specific and C. pecorum-specific LAMP assays that are able to detect these bacteria in clinical samples in either the laboratory or POC settings. With further development and a focus on the preparation of these assays at the POC, it is anticipated that both tests may fill an important niche in the repertoire of ancillary diagnostic tools available to clinicians.

Introduction

The obligatory intracellular bacteria, Chlamydia psittaci and Chlamydia pecorum, are globally widespread veterinary pathogens that cause disease in an astonishing range of hosts. C. psittaci, the causative agent of psittacosis or wasting bird disease, is regarded as a major economically relevant poultry and pet bird pathogen (Knittler & Sachse, 2015; Szymanska-Czerwinska & Niemczuk, 2016). Globally, C. psittaci infections are also sporadically reported in other animal species such as pigs, cattle, sheep and horses, resulting in asymptomatic shedding, acute respiratory disease and, in the case of horses, reproductive loss (Reinhold, Sachse & Kaltenboeck, 2011; Knittler & Sachse, 2015; Jelocnik et al., 2017). Importantly, this pathogen continues to pose risks to public health through zoonotic transmission events that may lead to severe pneumonia (Gaede et al., 2008; Laroucau et al., 2015; Branley et al., 2016). This zoonotic risk is typically associated with direct contact with C. psittaci infected birds, although indirect contact through exposure to environmental contamination has been suggested (Branley et al., 2014; Branley et al., 2016).

C. pecorum is perhaps best known as the major pathogen of the iconic Australian native species, the koala. These infections are most commonly asymptomatic but can also result in serious inflammatory ocular and/or urogenital disease, affecting almost all Australia’s mainland koala populations (Polkinghorne, Hanger & Timms, 2013; Gonzalez-Astudillo et al., 2017). C. pecorum is also an important livestock pathogen causing a range of debilitating diseases such as sporadic bovine encephalomyelitis, polyarthritis, pneumonia and conjunctivitis, with faecal shedding as a constant feature of these infections (Lenzko et al., 2011; Reinhold, Sachse & Kaltenboeck, 2011; Walker et al., 2015). In livestock, chlamydial pathogens such as C. pecorum and C. psittaci may be found as co-infections, raising the possibility of a synergistic pathogenic effect (Lenzko et al., 2011; Reinhold, Sachse & Kaltenboeck, 2011; Knittler & Sachse, 2015). The reports of chlamydial infections in novel hosts and their recognised pathogenic potential (Jelocnik et al., 2015b; Burnard & Polkinghorne, 2016; Taylor-Brown & Polkinghorne, 2017), further highlight the need for faster detection and molecular discrimination of infecting strains.

Whilst significant progress has been made in understanding the molecular epidemiology of C. psittaci and C. pecorum infections (Jelocnik et al., 2015a; Branley et al., 2016), the diagnosis and detection of these pathogens is still difficult, laborious and costly, challenging efforts to manage and treat infected hosts. A variety of traditional (cell culture, antigen detection, and serology) and molecular (conventional and real-time quantitative PCR (qPCR)) diagnostic options are used to detect chlamydial infections and diagnose chlamydiosis (Sachse et al., 2009). For both C. psittaci and C. pecorum, nucleic acid amplification tests (NAATs) are presently considered the diagnostic “gold standard” due to their specificity and sensitivity, however the use of these assays is mainly restricted to research and/or diagnostic laboratories. In the absence of standardised gene target(s) for these organisms, numerous single or nested species-specific qPCR assays have been proposed and/or are used for C. psittaci (Madico et al., 2000; Geens et al., 2005; Menard et al., 2006; Branley et al., 2008) and C. pecorum (Marsh et al., 2011; Higgins et al., 2012; Wan et al., 2011; Walker et al., 2016) diagnosis.

The development and use of low-cost molecular diagnostic tools performed at the point-of-care (POC) which fulfil the World Health Organization “ASSURED” criteria of affordable, sensitive, specific, user-friendly, rapid, equipment-free, and deliverable to those in need to be tested, are on the exponential rise (Maffert et al., 2017). While POC testing is not necessarily required when considering most chlamydial infections of veterinary concern, the ability to provide a rapid detection of infections becomes of increasing significance when veterinarians and other animal workers may be at risk of being exposed to C. psittaci infections in field or farm settings. POC testing is also particularly relevant for Chlamydia screening in wild animals where laboratory testing is not accessible either due to logistics associated with field sampling or that services are not routinely available for testing of samples from wildlife. The latter problem is particularly acute for diagnosing infections in koalas, with the recent decision to stop the production of a commercially viable solid-phase ELISA leaving wildlife hospitals unable to diagnose and successfully treat asymptomatic C. pecorum infections (Hanger et al., 2013).

While there are many options for molecular POC diagnostics, Loop Mediated Isothermal Amplification (LAMP) assays developed for use in pathogen diagnostics are popular as they offer significant advantages over PCR and/or serology testing (Maffert et al., 2017). Rapid, simple, highly specific, easy to interpret, and carried out at a constant temperature, LAMP assays can provide a diagnosis in 30 min, in either laboratory or field setting (Mansour et al., 2015; Notomi et al., 2015). Rapid isothermal LAMP assays that could be performed at the POC targeting human C. pneumoniae (Kawai et al., 2009) and C. trachomatis (Jevtusevskaja et al., 2016; Choopara et al., 2017) infections have been proposed for use in chlamydial diagnostics. Development of a C. pecorum LAMP, in particular, would meet immediate demand for koala C. pecorum infections diagnostics, providing an alternative solution for the current laboratory diagnostics. A recent outbreak of psittacosis in veterinary staff and students in contact with a C. psittaci-infected and sick neonatal foal (Chan et al., 2017; Jelocnik et al., 2017), further demonstrates the need for POC assays such as LAMP to rapidly diagnose C. psittaci. In the present study, we describe the development and evaluation of rapid and robust C. psittaci-specific and C. pecorum-specific LAMP assays for detection of these organisms in either laboratory or POC settings.

Materials and Methods

Bacterial cultures and clinical samples used in this study

C. psittaci LAMP assay was evaluated using: (1) 12 DNA samples extracted from previously characterised C. psittaci isolates (10 human, two parrot and one equine) (Table S1); (2) DNA extracted from 21 placental, foetal, nasal, lung and rectal swabs, and 1 each placental and foetal tissue sample taken from 20 equine hosts; and (3) three pigeon liver DNA extracts (Table S2). All samples were collected and submitted as part of routine diagnostic testing by field or district veterinarians to the State Veterinary Diagnostic Laboratory (SVDL), Elizabeth Macarthur Agricultural Institute (EMAI), Menangle, NSW, Australia, and as such do not require special animal ethics approval. DNA extracts from these samples were kindly provided by Dr. Cheryl Jenkins, and Dr. James Branley. The use of these swabs was considered by the University of The Sunshine Coast (USC) Animal Ethics Committee and the need for further ethics consideration was waived under exemption AN/E/17/19.

C. pecorum LAMP was evaluated using a: (1) 18 DNA samples extracted from previously characterised koala (n = 7), sheep (n = 4), cattle (n = 4) and pig C. pecorum (n = 3) cultures (Table S1); (2) 16 sheep and 13 cattle ocular, rectal, and tissue swab DNA samples; and (3) 34 ocular and urogenital (UGT) koala swab DNA samples (Table S3), all available in our collection. The use of these swabs, also collected by qualified veterinarians as a part of routine diagnostic testing, was considered and approved for exemption by the University of The Sunshine Coast (USC) Animal Ethics Committee (AN/E/14/01 and AN/E/14/31).

We also evaluated the specificity of the assays against DNA samples extracted from previously characterised (i) chlamydial isolates (koala C. pneumoniae LPColN, C. abortus S26/3, C. suis S45, C. trachomatis serovar D, C. murridarum Nigg, C. caviae GPIC) and uncultured Chlamydiales (Fritschea spp.); (ii) Gram negative Escherichia coli and Prevotella bivia; Gram positive Fusobacterium nucleatum, Staphylococcus epidermidis, S. aureus, Streptococcus spp., and Enterococcus faecalis; and (iii) commercially available human gDNA (Promega, Alexandria, NSW 2015), all available in our laboratory (Table S1).

In order to evaluate rapid swab processing, 18 ocular, cloacal and UGT (14 dry and four RNA-Later) clinical swabs taken from 14 koalas with presumptive chlamydiosis were used for testing without DNA extraction. Briefly, RNA-Later and dry swabs with added 500 µL TE buffer were vortexed vigorously for 5 min. 300 µL aliquots were then heated to 98 °C for 15 min to lyse DNA, following LAMP testing. The use of these swabs, collected as a part of routine diagnostic testing, is also under Animal Ethics approval exemption (AN/E/14/01). An aliquot of 50 µL of the swab suspension was used for LAMP and qPCR assays, while from the remaining volume of the swab suspension was used for DNA extraction, in order to compare swab suspension and its paired extracted DNA as a template in the assays.

LAMP assays design

For the C. psittaci-specific LAMP gene target, we targeted a previously described conserved single-copy C. psittaci-specific CDS, encoding for hypothetical protein and denoted Cpsit_0607 in the representative C. psittaci 6BC strain (Genbank accession number NC_015470.1) (Voigt, Schöfl & Saluz, 2012). This gene was also previously proposed as a target for molecular diagnosis of C. psittaci infections (Opota et al., 2015).

The C. pecorum-specific candidate LAMP gene target, encoding for a single-copy conserved hypothetical protein and denoted CpecG_0573 in the C. pecorum MC/Marsbar koala type strain (GenBank accession number NZ_CM002310.1), was selected based on a comparative genomics analysis of published koala and livestock C. pecorum genomes (Jelocnik et al., 2015a). For the purposes of this study, we will refer to it as Cpec_HP. Both candidate gene sequences were aligned to the corresponding allele from other publicly available C. psittaci or C. pecorum strains using Clustal X (as implemented in Geneious 9 (Kearse et al., 2012)), and analysed in blastn against the nucleotide collection nr/nt database to assess intra-species sequence identity, and inter-species specificity.

For C. ps_0607 alignment, besides 6BC, we used the gene alleles from strains 84/55 (CP003790.1), 02DC15 (CP002806.1), 01DC11 (CP002805.1), WC (CP003796.1), 01DC12 (HF545614.1), NJ1 (CP003798.1), CR009 (LZRX01000000), Ho Re upper (LZRE01000000) and PoAn (LZRG01000000). For C. pec _HP alignment, besides MC/Marsbar, we used the gene alleles from E58 (CP002608.1), P787 (CP004035.1), W73 (CP004034.1), IPA (NZ_CM002311.1), NSW/Bov/SBE (NZ_JWHE00000000.1), L71 (LFRL01000000), L17 (LFRK01000001), L1 (LFRH00000000), DBDeUG (NZ_CM002308.1), SA/K2/UGT (SRR1693792), Nar/S22/Rec (SRR1693794) and Mer/Ovi1/Jnt (SRR1693791).

Species-specific LAMP primers were designed using the target sequences with the open-source Primer Explorer v5 software (Eiken Chemical Co., Tokyo, Japan) and licensed LAMP Designer 1.15 software (Premier Biosoft, Palo Alto, CA, USA). For both C. pecorum and C. psittaci, Primer Explorer v5 yielded five sets of four LAMP primers including two outer (forward F3 and backward B3) primers and two inner (forward inner FIP and backward inner BIP) primers targeting different regions of the target gene, while LAMP Designer yielded single best set of six LAMP primers including two outer primers (forward F3 and backward B3), two inner primers (forward inner FIP and backward inner BIP) and two loop primers (forward loop LF and backwards loop LB). All primers (as single or paired) were tested in silico, including analysing primer sequences in blast for species specificity and OligoAnalyser 3.1 (available from http://sg.idtdna.com/calc/analyzer) for primer dimerization, hairpins and melting temperatures.

After in silico and in LAMP reaction testing, a set of four primers designed by PrimerExplorer v5 and targeting a 209 bp region of the C. pec_HP gene (spanning from position 22 to 230) was selected for C. pecorum LAMP assays performed in this study. Additional loop primers (LF/LB) were also designed to accelerate amplification time and increase sensitivity. For C. psittaci, a set of six primers designed with LAMP Designer and targeting a 263 bp region of the C. ps_0607 gene (spanning from position 286 to 548) was selected for LAMP assays performed in this study. The specificity of primer sequences was assessed in silico using discontiquousBLAST analyses. Amplicons generated by conventional PCR using outer F3 and B3 primers for both C. psittaci and C. pecorum were gel excised, purified using Roche High Pure purification kit, and sent to Australian Genome Research Facility (AGRF) for Sanger sequencing for sequence identity confirmation.

LAMP assay optimisation

Both C. psittaci and C. pecorum LAMP assays were carried out in a 25 µL reaction volume. The reaction mixture consisted of 15 µL Isothermal Master Mix ISO001 (Optigene, Horsham, UK), 5 µL six primers mix (at 0.2 µM F3 and B3, 0.8 µM FIP and BIP, and 0.4 µM LF and LB) and 5 µL template, following LAMP assay run at 65 °C in the Genie III real-time fluorometer (Optigene, Horsham, UK), as per manufacturer instructions. Following determination of the most optimal conditions (fastest amplification time, fluorescence and annealing temperature), C. psittaci LAMP assays were run at 65 °C for 30 min followed by annealing step of 98–80 °C at a rate of 0.05 °C/s, while C. pecorum LAMP assays were run using the same temperature and annealing conditions, however for 45 min. A negative control (LAMP mix only) was included in each run. Both C. psittaci and C. pecorum LAMP assays were performed on a thermal cycle heating block at 65 °C for 30 min, following detection of amplicons by electrophoresis on a 1.5% ethidium bromide agarose gel and visualisation under UV. In addition, several C. pecorum LAMP assays were conducted using the four primer set, two outer (F3 and B3) and two inner (FIP and BIP) primers, on a heating block at 65 °C for 45 min.

After the assay optimisation, LAMP testing was evaluated using previously tested clinical samples, previously characterised isolates and untested new samples. C. pecorum-presumptive samples were simultaneously tested using our in-house C. pecorum–specific qPCR assay (Marsh et al., 2011), while C. psittaci-presumptive samples were tested using a pan-Chlamydiales qPCR assay with primers 16SIGF and 16SIGR targeting the 298 bp 16S rRNA fragment (Everett, Bush & Andersen, 1999). Amplicon sequencing was used for the latter assay to confirm species identity. The qPCR assays were carried out in a 20 µL total volume, consisting of 10 µL SYBR™ Green PCR Master Mix (Life Technologies Australia Pty Ltd., Scoresby, Victoria, Australia), 1 µL of each 10 µM forward and reverse primer, 3 µL miliqH2O, and 5 µL DNA template. The qPCR assays were run for 35 cycles (Ct), and in each qPCR assay a positive (cultured C. pecorum and/or C. psittaci DNA) and negative (miliqH2O) controls were included. Based on the qPCR standard curve and the number of running cycles, samples amplifying at >30 Ct (and/or equivalent detected genome copy number) were considered negative. The 23 C. psittaci-presumptive equine samples were also tested with a C. psittaci-specific qPCR assay targeting the 16S rRNA gene/16S-23S rRNA spacer gene (Madico et al., 2000) at the State Veterinary Diagnostic Laboratory (SVDL), Elizabeth Macarthur Agricultural Institute (EMAI), Menangle, NSW, Australia. Samples amplifying at >39 Ct were considered negative. LAMP testing was performed in a blind fashion, by two different operators, unaware of qPCR results.

Statistical analyses

For each assay, we compared the performance of two tests evaluated in the same population by calculating Kappa and overall agreement, as well as estimated sensitivity and specificity (with specified Clopper–Pearson (exact) confidence limits) of LAMP compared to the known reference (gold standard) qPCR test using EpiTools online (Sergeant, 2017). It is suggested the Kappa value be interpreted as follows: values ≤0 as indicating no agreement and 0.01–0.20 as none to slight, 0.21–0.40 as fair, 0.41–0.60 as moderate, 0.61–0.80 as substantial, and 0.81–1.00 as almost perfect agreement.

Results and Discussion

With the emergence of new spill-over threats posed by C. psittaci (Laroucau et al., 2015; Jelocnik et al., 2017), there is an increasing need for rapid diagnostic tools for this pathogen, particularly for those that may have practical application in the field or clinical setting. There are specific needs for C. pecorum POC tests as well in both the veterinary care and treatment of infected domesticated and native animals, particularly in settings where veterinary diagnostic testing is logistically challenging. In the present study, to the best of our knowledge, we describe the first development of novel, rapid and robust C. psittaci-specific and C. pecorum-specific LAMP assays that are able to detect these bacteria in clinical samples in either the laboratory or POC settings.

C. psittaci and C. pecorum LAMP development

A C. psittaci-specific gene (C.ps_0607) was previously characterised as a conserved gene sequence present only in C. psittaci genomes, and absent from all other related chlamydial species (Voigt, Schöfl & Saluz, 2012). BLAST analyses and alignment of the C.ps_0607 gene sequences, including those from recently described human, bird and equine Australian isolates, confirmed species specificity and sequence conservation. Between 0 and 13 single nucleotide polymorphisms (SNPs) were observed amongst strains (100–95.1% sequence identity) based on a 263 bp alignment of C.ps_0607 gene sequences, including that from the most distant C. psittaci NJ1 taxon (Fig. S1A). Similarly, the C. pecorum HP gene (denoted CpecG_0573 locus in Marsbar strain) was determined as a highly conserved species-specific sequence following BLAST analysis against publicly available sequences. Using an alignment of HP gene sequences from 14 publicly available C. pecorum genomes, there were only two SNPs in the 209 bp region to be targeted by LAMP (Fig. S1B).

Although multiple LAMP primer sets were predicted, LAMP primer sets denoted in Fig. 1 were chosen for further assay development. For C. psittaci assays, a set designed using LAMP Explorer was utilised while, for C. pecorum, we used a set designed with PrimerExplorer (Table 1). After initial testing, some of the predicted primer sets were discarded due to (i) potential cross-amplification associated with a lack of specificity of the target primer; (ii) not achieving an amplification signal in the fluorometer; and (iii) amplifying non-specific targets, including positive amplification in negative controls (data not shown). While we achieved initial amplification of a C. psittaci single copy dilution in a 30 min assay using the designed LAMP primer set, initial reaction times for a C. pecorum single copy amplification averaged 50 min. In order to accelerate amplification times for C. pecorum, we additionally designed a pair of Loop primers for the C. pecorum set which decreased the amplification of a single copy to 30 min.

Figure 1 LAMP primer sequences and positions in the target gene regions.

(A) C. psittaci LAMP primer set; and (B) C. pecorum LAMP primer set. Outer F3 and B3 primers are indicated in green, inner FIP and BIP in blue, and loop LF and BL in pink colour.

Table 1 LAMP primers set used in this study.

Name	Sequence 5′–3′	Position	Length	
C. psittaciLAMP primers	
F3	AGAACCGGATTAGGAGTCTT	286	20	
B3	GCTGCTAAAGCGAGTATTGA	548	20	
FIP(F1c + F2)	TCCGCAGTTTGTTCCATCACCCAA GGGTTATTCGACAACTACT		43	
BIP(B1c + B2)	ACTATGGATCGGCCACACATGGG TATGTTGCTTTGAATGGG		41	
LoopF	TTCAGGTAATCACGCACTTGA	350	21	
LoopB	TTCCCCACACTATTAAACAGCA	431	22	
F2	CAAGGGTTATTCGACAACTACT	307	22	
F1c	TCCGCAGTTTGTTCCATCACC	387	21	
B2	GGTATGTTGCTTTGAATGGG	472	20	
B1c	ACTATGGATCGGCCACACATG	410	21	
C. pecorumLAMP primers	
F3	ATCGGGACCTTCTCATCG	22	18	
B3	GCTGTTGTAAGGAAGACTCC	230	20	
FIP(F1c + F2)	GACTAACAGTATAAGCAGTGCTG TTAGTCTGCTGTCCAACTACA		44	
BIP(B1c + B2)	TTATCTCTCGTTGCAATGATAGGAG CCAACAGGATCAAACCAACTT		46	
LoopF	CTGAATTCGTTGAC	93	14	
LoopB	TACTGTCTTCACC	165	12	
F2	AGTCTGCTGTCCAACTACA	47	19	
F1c	GACTAACAGTATAAGCAGTGCTGTT	129	25	
B2	CAACAGGATCAAACCAACTT	210	20	
B1c	TTATCTCTCGTTGCAATGATAGGAGC	130	26	

Species-specificity for C. psittaci and C. pecorum LAMP targets was tested in the developed LAMP assays using DNA extracts from 12 C. psittaci and 18 C. pecorum cultured isolates, DNA extracts from other chlamydial species and a range of DNA extracts from other bacteria. Positive amplification as assessed by the presence of an observable amplification curve characterised by a specific melt was observed only for the target species in their respective assays (Table S1). No amplification curves were observed for any of the non-targeted chlamydial species or other bacteria included in our specificity assays (Table S1). The C. pecorum and C. psittaci LAMP assays did not amplify either the related chlamydial species or other bacteria included in our specificity assays. In this study, in contrast, a previously described “C. pecorum-specific” qPCR assay (Marsh et al., 2011; Wan et al., 2011) showed positive amplification and melt for C. psittaci and C. pneumoniae DNA samples.

The choice to use the C. ps_0607 gene as a LAMP target was straight forward since it had been suggested for such a purpose in previous studies (Voigt, Schöfl & Saluz, 2012; Opota et al., 2015), For C. pecorum, however, we utilised our ongoing comparative genomics to select C. pecorum-specific and conserved C.pec_HP gene described in this study for the first time. In silico analyses and assay development confirmed species-specificity of this gene and its suitability for use in diagnostic assays. Previously published C. pecorum diagnostic assays targeted highly polymorphic genes such as ompA (Higgins et al., 2012; Yang et al., 2014), which may require the use of probes due to sequence variation, prolonging the detection time and increasing diagnostic costs. Our routinely used in house C. pecorum-specific assay which targets a 204 bp 16S rRNA fragment (Marsh et al., 2011; Wan et al., 2011) was simpler to use, however we have shown that this assay may cross-react with other related chlamydial species due to a lack of sufficient sequence variation in the region of the 16S rRNA gene targeted (Bachmann, Polkinghorne & Timms, 2014). For koala diagnostics where C. pecorum is the most abundant and prevalent chlamydial organism (Polkinghorne, Hanger & Timms, 2013), this cross-reactivity may not be of a big concern. For the veterinary diagnosis of infections in livestock where co-infections with several chlamydial species are common (Lenzko et al., 2011; Reinhold, Sachse & Kaltenboeck, 2011), this assay may be less suitable. Using the C. pecorum-specific HP gene as a target in different diagnostic assays would hence seem promising.

Performance of the C. psittaci and C. pecorum LAMP assays

The sensitivity of the LAMP assays was evaluated using 5 µL cultured C. psittaci and C. pecorum gDNA in 10-fold serial dilutions as a template in assays performed in triplicate in separate runs. The limits of detection of the LAMP assays were conservatively 10 copies for C. psittaci, with 3/3 (100%) positive amplification for 10 copy dilutions for C. psittaci, and one copy for C. pecorum, with 3/3 (100%) positive amplifications for a single copy dilution of C. pecorum DNA (Tables 2 and 3). In the final and optimised LAMP assays, the mean amplification time detecting the lower limit (a single copy) for C. psittaci was 14.23 min with an average 84.45 °C melt (Table 2) while, for C. pecorum, it was 24 min with an average 83.42 °C melt (Table 3). Comparing the two newly developed assays, C. psittaci LAMP had the faster run time than that of C. pecorum LAMP. This difference in assays kinetics could be attributed to the improved C. psittaci LAMP primers design, as they were predicted by the LAMP Designer software (Nagamine, Hase & Notomi, 2002). As we additionally designed Loop primers for C. pecorum, we can anticipate an improvement in the C. pecorum assay kinetics by re-designing the loop primers (e.g., extending the sequence to 20–22 bp), as well as testing LAMP mixes in different ratios and with improved polymerases.

Table 2 C. psittaci LAMP assaya sensitivity.

Dilutiond	Time to amplify (min)	Melt (°C)	Time (Mean + SD)	Melt (Mean + SD)	
106	5.15	84.43	5.10, 0.09	84.49, 0.08	
106	5.00	84.46	
106	5.15	84.58	
105	6.30	84.34	6.30, 0.15	84.37, 0.06	
105	6.45	84.33	
105	6.15	84.43	
104	7.15	84.59	7.25, 0.09	84.56, 0.04	
104	7.30	84.58	
104	7.30	84.51	
103	8.45	84.46	8.25, 0.173	84.44, 0.01	
103	8.15	84.43	
103	8.15	84.44	
100	9.15	84.48	9.30, 0.15	84.46, 0.06	
100	9.30	84.39	
100	9.45	84.51	
10	12.00	84.41	11.33, 0.58	84.38, 0.03	
10	11.00	84.35	
10	11.00	84.39	
1	16.00	84.44	14.23, 2.51	84.34, 0.14	
1	0.00	0	
1	12.45	84.24	
0.1	25.25	84.20			
0.1	–b	–	–	84.20	
0.1	–	84.20c			
Notes.

a The assay was performed in Genie III Real-time fluorometer, with the amplification times and annealing temperatures recorded at the end of each run. The samples were tested in three different runs.

b No amplification detected.

c No amplification, but melt and annealing curve recorded.

d Template was serially diluted C. psittaci CR009 gDNA which genome copy number was determined by qPCR.

Table 3 C. pecorum LAMP assaye sensitivity.

Dilutionf	Time to amplify (min)	Melt (°C)	Time (Mean + SD)	Melt (Mean + SD)	
107ka	10.00	83.23	10.23, 0.32	83.30, 0.1	
107k	10.45	83.37	
106k	13.15	83.57	12.92, 0.40	83.51, 0.16	
106sb	13.15	83.33	
106cc	12.45	83.62	
105k	14.00	83.52	14.10, 0.17	83.48, 0.11	
105s	14.00	83.35	
105c	14.30	83.57	
104k	15.45	83.56	16.30, 0.78	83.44, 0.11	
104s	16.45	83.33	
104c	17.00	83.42	
103k	19.00	83.50	18.87, 1.35	83.45, 0.06	
103s	17.45	83.39	
103c	20.15	83.47	
100k	20.15	83.47	20.35, 2.00	83.33, 0.21	
100s	18.45	83.09	
100c	22.45	83.42	
10k	22.30	83.52	22.43, 1.50	83.42, 0.1	
10s	21.00	83.42	
10c	24.00	83.33	
1k	23.15	83.52	23.92, 2.11	83.41, 0.12	
1s	22.30	83.42	
1c	26.30	83.28	
0.1k	36.00	83.41	34.65, 1.91	83.39, 0.06	
0.1s	–d	83.43	
0.1c	33.30	83.33	
Notes.

a Koala Marsbar isolate.

b Sheep IPA isolate.

c Cattle E58 isolate.

d No amplification, but melt and annealing curve recorded.

e The assay was performed in Genie III Real-time fluorometer, with the amplification times and annealing temperatures recorded at the end of each run. The samples were tested in different runs.

f Template was serially diluted C. pecorum gDNA which genome copy number was determined by qPCR.

In order to test the reproducibility of our LAMP assays, we tested a subset of C. pecorum and C. psittaci PCR positive samples (Table 4). All samples were run in a “blind fashion”, in triplicate and in separate runs by two different operators. For both assays, the amplification times and melts of each sample between the runs were very similar, with 0 to 1.5 min (SDs ranging from 0–0.98) difference in amplification times for each sample, and 0.03 to 0.83 °C (SDs ranging from 0.02–0.26) difference in melt for each sample. Congruence between the runs performed by different operators indicates that both LAMP assays described in this study are highly reproducible, and can detect the target organism in less than 30 min even when in low infectious loads of <10 copies.

Table 4 Reproducibility of the LAMP testing using clinical and cultured samples.

Samples	Runa	Time to amplify (min)	Melt (°C)	Time (Mean + SD)	Melt (Mean + SD)	
C. pecorumpositive samples	
Koala rectal swab	1	20.15	83.44	20.53, 0.54	83.32, 0.16	
2	20.30	83.37	
3	21.15	83.14	
Marsbar DNA	1	13.50	83.50	13.27, 0.20	83.55, 0.06	
2	13.15	83.52	
3	13.15	83.62	
Koala A2 DNA	1	12.00	83.35	11.43, 0.51	83.41, 0.05	
2	11.00	83.45	
3	11.30	83.43	
RI koala UGT swab	1	17.00	83.34	17.72, 0.62	83.21, 0.12	
2	18.00	83.21	
3	18.15	83.09	
L14 DNA	1	13.15	83.53	13.15, 0	83.50, 0.02	
2	13.15	83.50	
3	13.15	83.48	
HsLuRz DNA	1	13.45	83.49	13.63, 0.32	83.40, 0.08	
2	13.45	83.36	
3	14.00	83.34	
K20 cloaca swab	1	22.00	82.83	22.2, 0.23	83.01, 0.19	
2	22.15	83.00	
3	22.45	83.20	
C. psittacipositive samples	
Cr009 DNA	1	6.45	84.30	6.40, 0.09	84.33, 0.03	
2	6.45	84.36	
3	6.30	84.34	
HoRE DNA	1	5.00	84.46	5.10, 0.08	84.45, 0.14	
2	5.15	84.58	
3	5.15	84.30	
B2 DNA	1	10.30	84.08	10.10, 0.17	84.17, 0.08	
2	10.00	84.20	
3	10.00	84.24	
Horse placental swab	1	11.15	82.90	10.58, 0.49	83.19, 0.26	
2	10.30	83.42	
3	10.30	83.24	
Horse_pl DNA	1	10.30	84.53	10.87, 0.98	84.41, 0.18	
2	12.00	84.21	
3	10.30	84.50	
Notes.

a The assay was performed in Genie III Real-time fluorometer, with the amplification times and annealing temperatures recorded at the end of each run. The samples were tested in three different runs by two different operators.

Pathogen detection in clinical samples using newly developed LAMP

For C. psittaci, a total of 26 DNA extracts from clinical samples were tested with both C. psittaci LAMP and qPCR assays (Table S2). For these analyses, samples with >20 min amplification time were considered negative for LAMP, while for qPCR, samples with <20 genome copy/reaction and/or >30 Ct (quantification cycle) were considered negative, based on the qPCR standard curve and the number of running cycles used for this testing. As observed in Table S2 and based on above cut-off values, 24/26 (92.3%) samples were congruent between the two tests, with 11 samples positive and 13 samples negative by both (Table 5). For 2/26 (7.7%) where there was disagreement, one sample was LAMP positive but qPCR negative, and another was qPCR positive but C. psittaci LAMP negative. Based on these results, the Kappa value was calculated at 0.85 (95% CI [0.64–1.05]) indicating an almost perfect agreement between the tests. The overall sensitivity of the C. psittaci LAMP was 91.7% (Clopper–Pearson 95% CI [0.62–0.99]) and with 92.9% (Clopper–Pearson 95% CI [0.66–0.99]) specificity, compared to the qPCR used in this study. In addition, a subset of 23 samples was also tested independently by a third party. Using a cut off of >Ct 39 as negative, 19/23 (82.60 %) of these test results were in congruence with our C. psittaci LAMP results (Table S2).

Table 5 Comparison of the C. psittaci LAMP and qPCR methods for the organism detection in clinical samples.

Test	qPCR +ve	qPCR −ve	qPCR Total	
LAMP +ve	11	1	12	
LAMP −ve	1	13	14	
LAMP Total	12	14	26	

For C. pecorum, we tested a total of 63 DNA extracts from clinical samples from several animal hosts by both LAMP and qPCR (Table S3). For these analyses, samples with >30 min amplification time were considered negative for LAMP, while for qPCR, samples with <35 genome copy /reaction and/or >30 Ct were considered negative based on the standard curve and number of run cycles used for this testing. For the 63 clinical samples, the overall congruence was 84.1% with a Kappa value of 0.68 (95% CI [0.50–0.86]), indicating substantial agreement between the tests. Congruent results between tests were obtained for 53 samples, while there were 10 discrepant samples using the above cut off for C. pecorum (Table 6). The overall sensitivity of C. pecorum LAMP was 90.6 % (Clopper–Pearson 95% CI [0.75–0.98]), while specificity was 77.4 % (Clopper–Pearson 95% CI [0.59–0.90]) in comparison to the qPCR assay. A subset of 36 C. pecorum samples was also tested in a thermal cycler using the newly developed LAMP and results were determined as an end point detection. For this experiment, 34/36 (94.4%) samples were congruent between LAMP performed in fluorometer and in a thermal cycler (Table S3), demonstrating the robustness of the C. pecorum LAMP (Fig. S2).

Table 6 Comparison of the C. pecorum LAMP and qPCR methods for the organism detection in clinical samples.

Test	16s +ve	16s −ve	16s Total	
LAMP +ve	29	7	36	
LAMP −ve	3	24	27	
LAMP Total	32	31	63	

Table 7 Comparison of C. pecorum LAMP and qPCR for organism detection using rapidly processed swab samples and their DNA extracts.

Sample	LAMPa result for swab suspension	qPCRb result for swab suspension	LAMP result for DNA extract	qPCR result for DNA extract	LAMP result for “spiked” swab suspension	LAMP result for “spiked” DNA extract	qPCR result for “spiked” swab suspension	qPCR result for “spiked” DNA extract	
K1 ocularc	NEG	NEG	0.00/83.49	NEG	NEG	–	NEG	–	
K6 ocularc	NEG	NEG	21.00/83.23	3 × 103 (Ct 20)	NEG	–	NEG	–	
K9 ocularc	NEG	NEG	25.45/83.39	287 (Ct 24)	NEG	–	NEG	–	
K2 ocularc	NEG	NEG	NEG	NEG	NEG	–	NEG	–	
R1 eye	25.45/83.39	222 (Ct 25)	20.15/83.27	750 (Ct 24)	–	–	–	–	
R1 cloaca	30.00/83.34	NEG	NEG	NEG	11.15/83.47	12.15/83.42	5 × 103 (Ct 17)	1.5 × 103 (Ct 18)	
K eye	27.00/83.15	NEG	0.00/83.35	NEG	–	–	–	–	
Koala 2 eye	NEG	NEG	NEG	NEG	11.00/83.51	11.00/83.40	1.2 × 103 (Ct 19)	1.1 × 104 (Ct 15)	
Koala 2 cloaca	27.30/83.77	116 (Ct 26)	21.30/83.49	375 (Ct 25)	–	–	–	–	
Will Cloaca	0.00/83.77	NEG	NEG	NEG	12.00/83.45	11.00/83.34	1.5 × 103 (Ct 19)	8 × 103 (Ct 17)	
23117 Eye	21.30/83.20	NEG	23.15/83.23	150 (Ct 25)	–	–	–	–	
23117 Cloaca	22.00/83.29	NEG	24.00/83.15	90 (Ct 27)	–	–	–	–	
Flyn eye	NEG	NEG	NEG	NEG	12.30/83.50	11.00/83.35	1.9 × 103 (Ct 18)	8.3 × 103 (Ct 16)	
Tyke eye	NEG	NEG	NEG	NEG	12.00/83.44	10.45/83.40	1.3 × 103 (Ct 19)	9 × 103 (Ct 16)	
Bill eye	NEG	NEG	NEG	NEG	12.15/83.49	10.45/83.34	1.2 × 103 (Ct 19)	1 × 104 (Ct 15)	
Ray eye	NEG	NEG	NEG	NEG	12.45/83.49	11.00/83.40	4.7 × 103 (Ct 17)	1 × 104 (Ct 15)	
Ray cloaca	NEG	NEG	NEG	NEG	12.15/83.43	11.00/83.30	700 (Ct 20)	9 × 103 (Ct 16)	
Koala F Eye	NEG	NEG	NEG	NEG	11.45/83.45	11.00/83.35	1.3 × 103 (Ct 19)	1.1 × 104 (Ct 15)	
Notes.

a LAMP results are expressed as time to amplify (min) and melt (°C).

b qPCR results are expressed as copies/reaction and Ct value.

c RNA Later swabs.

Considering that the qPCR assay used in this study to quantify and detect C. psittaci is chlamydial genus rather species specific (Everett, Bush & Andersen, 1999), high congruence observed for C. psittaci assays could be attributed to testing a limited set of samples taken from hosts with presumptive C. psittaci chlamydiosis. Lower congruence between the C. pecorum-specific assays could be due to technical and experimental aspects and characteristics (such as the assay efficiency, analytical sensitivity, template preparation) (Bustin et al., 2010) of the C. pecorum 16S qPCR assay used in this study. As a sidenote, we also evaluated the use of C. psittaci and C. pecorum LAMP targets (263 bp of the C. ps_0607 and 209 bp C. pec_HP genes, respectively) using outer F3 and B3 primers in a fluorescence-based (SybrGreen) qPCR assays, if needed to estimate infectious loads of the pathogen. In this preliminary analyses, both targets seem suitable for use in qPCR assays as well, as we were able to detect low infectious load up to 10 copies/reaction in a sample.

Rapid swab processing

Rapid swab processing and using the swab suspension directly in LAMP assays were previously successfully evaluated for testing for respiratory syncytial virus from nasopharyngeal swabs (Mahony et al., 2013) and rapid detection of Streptococcus agalactiae in vaginal swabs (McKenna et al., 2017). A recent study also demonstrated that C. trachomatis can be detected directly from urine samples using the LAMP method (Jevtusevskaja et al., 2016). In this study, we also evaluated rapid swab processing without DNA extraction in order to begin to assess the POC potential of these assays. A total of 18 swabs taken from conjunctival and urogenital sites from koalas with presumptive chlamydiosis, of which four were stored in RNA Later and 14 were dry, were used for this experiment (Table 7). Vigorously vortexed and heated swab suspension samples were directly used as a template in both C. pecorum LAMP reaction performed in fluorometer and qPCR assay. We also performed DNA extraction from the swabs to be used as a comparison to rapid swab processing. We did not detect C. pecorum DNA in any of the RNA Later suspensions either by LAMP nor qPCR assay (Table 7), in contrast to detecting C. pecorum in 50% (2/4) of the DNA extracts from the swabs by both methods. Using the rapidly processed swab suspension as a template, C. pecorum was detected in 6/14 by LAMP, and only 2/14 by qPCR (Table 7). The swab suspension LAMP results were 92.8% (13/14) congruent to the LAMP results and 85.7% congruent (12/14) to the qPCR results using the swab’s paired DNA sample. In order to evaluate the potential presence of inhibitors in our samples, we “spiked” negative swab suspensions and its paired DNA samples with known amounts of C. pecorum (1 × 104 copies/reaction). As observed in Table 7, we detected C. pecorum by both LAMP and qPCR in “spiked” negative samples derived from dry swabs. No C. pecorum was detected in “spiked” RNA Later swab suspension, indicating the potential presence of inhibitors in these reactions. Our results suggest that the LAMP assays are capable of amplifying specific amplification products from crude DNA extracts.

Further work is additionally required to enhance the POC capabilities of these new chlamydial LAMP assays to meet the clinical need including (i) the evaluation of rapid swab processing methods using commercially available DNA release portable devices and/or sample preparation using microfluidic support; (ii) alternative amplification detection methods such as visible colorimetric or turbidimetric change and/or solid-phase ‘dipstick’ tests (Maffert et al., 2017). With further development and the aforementioned focus on the preparation of these assays at the POC (Parida et al., 2008; Tomita et al., 2008), it is anticipated that both LAMP tests described in this study may fill an important niche in the repertoire of ancillary diagnostic tools available to clinicians.

Supplemental Information

Figure S1 C. psittaci and C. pecorum LAMP targets sequences alignment

A: A 263bp fragment of the Cps_0607 gene alignment; B: A 209bp fragment of the C.pec_HP gene alignment. SNPs in the alignments are highlighted.

Click here for additional data file.

Figure S2 End-point detection of C. pecorum and C. psittaci LAMP performed on the thermal cycle

(A) C. pecorum LAMP sensitivity. In lanes 1: DNA marker VIII, 2: C. pecorum 105 genome copy number dilution, 3: 104 genome copy number dilution, 4: 103 genome copy number dilution, 5: 102 genome copy number dilution, 6: 101 genome copy number dilution, 7: 100 genome copy number dilution, 8: 10−1 genome copy number dilution, 9: 10−2 genome copy number dilution, 10: Negative (water as template), 11: Negative (mix only); (B) C. pecorum LAMP testing of clinical samples. In lanes 1: koala swab sample Chloe U, 2: Chloe R, 3: koala swab sample K61528, 4: K68199, 5: K67797, 6: K52866, 7: K60652, 8 and 11: DNA Marker VIII, 9: C. pecorum koala Marsbar strain, 10: Negative (water), 12: sheep eye swab S87, 13: Lamb1 eye swab, 14: cattle rectal swab 18R, 15: Cow brain sample, and 16: cattle rectal swab 20R; (C) C. psittaci LAMP sensitivity. In lanes 1: DNA marker VIII, 2: C. psittaci 105 genome copy number dilution, 3: 104 genome copy number dilution, 4: 103 genome copy number dilution, 5: 102 genome copy number dilution, 6: 101 genome copy number dilution, 7: 100 genome copy number dilution, 8: 10−1 genome copy number dilution, 9: 10−2 genome copy number dilution, 10: Negative (water as template), 11: Negative (mix only); (D) C. psittaci LAMP testing of clinical samples. In lanes 1: empty, 2: DNA Marker VIII, 3: Horse 14092/1, 4: Horse 10272/3, 5: Horse 13234/3, 6: Horse 11310/2, 7: Horse 12004/2, 8: Horse 12818/3, 9: Horse 11786/3, 10: Horse 11035/1, 11: Horse 13237/3, 12: Pigeon p12, 13: C. psittaci B2, 14: Negative (water as template); (E) C. pecorum LAMP specificity. In lanes 1: C. psittaci, 2: C. pneumoniae, 3: C. abortus, 4: C. suis, 5: C. trachomatis, 6: Chlamydiales positive sample (Fritchea spp.), 7 and 10: DNA Marker VIII, 8: C. pecorum koala Marsbar strain, 9: C. pecorum E58 cattle strain, 11 - 16: Gram Neg and Gram Pos bacteria, as follows: Escherichia coli, Enterococcus feacalis, Fusobacterium spp., Prevotella bivia, Staphylococcus epidermidis, Streptococcus spp., 17: empty, and 18-19: Negative (water and mix only).

Click here for additional data file.

Table S1 Specificity of the C. psittaci and C. pecorum LAMP assays

Click here for additional data file.

Table S2 C. psittaci LAMP testing of clinical samples

Click here for additional data file.

Table S3 C. pecorum LAMP testing of clinical samples

Click here for additional data file.

We thank Prof. James Mahony, Dr. Catherine Chicken, Dr. Joan Carrick, Dr. Ian Marsh, Narelle Sales and Dr. Bill Lott for their helpful advice on POC assays. We also thank Dr. Brendon O’Rourke, Sankhya Bommana, Sharon Nyari, Noa Ziklo and Alyce Taylor-Brown for provision of DNA samples used in this study. We also thank Sean McDonald, Geneworks, Australia, for providing us with the Genie III Fluorometer.

Additional Information and Declarations

Competing Interests

Author Contributions

Animal Ethics

Data Availability

The authors declare there are no competing interests.

Martina Jelocnik conceived and designed the experiments, performed the experiments, analyzed the data, contributed reagents/materials/analysis tools, wrote the paper, prepared figures and/or tables, reviewed drafts of the paper.

Md. Mominul Islam, Danielle Madden, Cheryl Jenkins and Scott Carver performed the experiments, analyzed the data, reviewed drafts of the paper.

James Branley performed the experiments, reviewed drafts of the paper.

Adam Polkinghorne analyzed the data, contributed reagents/materials/analysis tools, reviewed drafts of the paper.

The following information was supplied relating to ethical approvals (i.e., approving body and any reference numbers):

The use of these swabs was considered by the University of The Sunshine Coast (USC) Animal Ethics Committee and the need for further ethics consideration was waived under exemption AN/E/17/19.

The following information was supplied regarding data availability:

The raw data has been supplied as Supplementary Files.

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
