# Peer review of "Development and evaluation of rapid novel isothermal amplification assays for important veterinary pathogens: Chlamydia psittaci and Chlamydia pecorum"

_PeerJ, doi:10.7717/peerj.3799_

## Round 0.1 · original submission · Minor Revisions

This is a well-written and well-designed study with clinical relevance, as the targeted Chlamydia species are important veterinary pathogens. The developed LAMP assay stands to benefit the community interested in combating these serious pathogens. Please pay specific attention to the concerns of the reviewers, particularly reviewer 1. Consider the electrophoresis experiment to make the approach more practical across labs that may not have access to a real-time fluorometer. I believe that addressing all of the concerns of the reviewers will suffice for publication in PeerJ. Well done, and good luck with your revision!

Reviewer 1 ·

Basic reporting

Chlamydia psittaci and C.pecorum are impotant veterinary pathogens with great zoonotic potentials. Jelocnik et al. established a specific and sensitive LAMP method for their detection. This manuscript is extremely well written, and the conclusions are well supported by the performed assays.

Experimental design

This work was well designed by validating the method from different angels, comparing to the existing assay, and testing on different samples; Also, sufficient types of positive and negative controls have been used.

Validity of the findings

The data in this MS is statistically sound;
The conclusion made in this MS is solid;

Additional comments

This reviewer has following comments regarding this manuscript, and the authors may consider to address them.
1) This reviewer sincerely appreciates the efforts of the authors to test the sensitivity and specificity of the established assays with different approaches. However, a spiking assay would make everything more meaningful, instead of testing on the pure isolates or on the clinical samples separately. The authors may consider to spike serially diluted C. psittaci or C. pecorum to the samples (negative swabs); The spiked samples will be tested by your assays and existing PCRs simultaneously until you find the breakpoints. This is realistic question when you apply the established assays in field samples, and during this process both efficiency of DNA extraction and PCR amplification are considered;

2) In this assay, the real-time fluorometer was used while this machine is not available in many labs. For this purpose and part of the validation of a new assay, the authors should provide a nice image of gel elctrophoresis on the LAMP products (showing the sensitivity);

3) The figure 1 needs to be modified. The regions outside of the primers should be left out, making the figure easier to read;

4) Table 7 is confusing to this reviewer;

5) Lines 149-155 indicate that other chlamydial species were used to validate the assays; however, not much was mentioned in Results (Only C. pneumoniae). This is very important, one of the advantages of this assay over other LAMP assay. The authors may indicate here that any copy number of other chlamydial species cannot be amplified by the assays; or copy number below certain number turned to be negative in this assay.

·

Basic reporting

The authors clearly described the development and assay evaluations of C. psittaci-specific and C. pecorum-specific LAMP. The authors use professional English language throughout. The research is self-contained to the hypotheses and the finding methods are important to the animal health diagnostics. However, minor comments and advices are noted along the annotated pdf file.

Experimental design

Research question is well defined, and experimental design is appropriate and sufficient. If possible, a comparison of results to a commercially viable solid-phase ELISA may be included, and the authors should discuss the reason for discontinued use of the commercially viable solid-phase ELISA and its pros and cons compared with the newly developed LAMP. Additional notes are in the annotated pdf.

Validity of the findings

Data is clear and robust. Conclusion is well stated.

Additional comments

The article is well written.

---

## Round 0.2 · accepted · Accept

Dear Dr. Jelocnik and colleagues:
Thank you for addressing the careful and thorough reviews of your work. It is my opinion that you successfully addressed the concerns of both reviewers and made your work much better, as well as ready for publication in PeerJ. Congratulations! This will undoubtedly be an important contribution to veterinary research on chlamydial pathogens. Well done! I look forward to the final product!
-joe